# The N-Terminal Part of the 1A Domain of Desmin Is a Hot Spot Region for Putative Pathogenic *DES* Mutations Affecting Filament Assembly

**DOI:** 10.3390/cells11233906

**Published:** 2022-12-02

**Authors:** Andreas Brodehl, Stephanie Holler, Jan Gummert, Hendrik Milting

**Affiliations:** Erich and Hanna Klessmann Institute, Heart and Diabetes Center NRW, University Hospital of the Ruhr-University Bochum, 32545 Bad Oeynhausen, Germany

**Keywords:** desmin, myopathy, cardiomyopathy, intermediate filaments, cytoskeleton, myofibrillar myopathy (MFM), desminopathy, desmosomes, protein aggregation

## Abstract

Desmin is the major intermediate filament protein of all three muscle cell types, and connects different cell organelles and multi-protein complexes such as the cardiac desmosomes. Several pathogenic mutations in the *DES* gene cause different skeletal and cardiac myopathies. However, the significance of the majority of *DES* missense variants is currently unknown, since functional data are lacking. To determine whether desmin missense mutations within the highly conserved 1A coil domain cause a filament assembly defect, we generated a set of variants with unknown significance and systematically analyzed the filament assembly using confocal microscopy in transfected SW-13, H9c2 cells and cardiomyocytes derived from induced pluripotent stem cells. We found that mutations in the N-terminal part of the 1A coil domain affect filament assembly, leading to cytoplasmic desmin aggregation. In contrast, mutant desmin in the C-terminal part of the 1A coil domain forms filamentous structures comparable to wild-type desmin. Our findings suggest that the N-terminal part of the 1A coil domain is a hot spot for pathogenic desmin mutations, which affect desmin filament assembly. This study may have relevance for the genetic counselling of patients carrying variants in the 1A coil domain of the *DES* gene.

## 1. Introduction

Desminopathies manifest clinically as different cardiomyopathies and/or skeletal myopathies [1,2]. The clinical spectrum of desminopathies is wide and heterogenous, and includes different skeletal myopathies and cardiomyopathies such as dilated (DCM) [3,4,5], arrhythmogenic (ACM) [6,7], hypertrophic (HCM) [8,9], restrictive (RCM) [10,11] and non-compaction cardiomyopathy (NCCM) [12,13]. Some patients develop a combined cardiac and skeletal muscle phenotype, and even within the same family different phenotypes can be present [14].

Genetically, desminopathies are caused by *DES* mutations [15,16]. The human *DES* gene (MIM, *125660) consists of nine exons localized on chromosome 2 [17] and encodes the major muscle-specific intermediate filament (IF) protein desmin. The majority of known pathogenic *DES* mutations are heterozygous missense or small in-frame deletion mutations. Of note, nonsense or frameshift mutations in the *DES* gene are rare and cause, in most cases, only in the homozygous or compound heterozygous status the phenotype [18,19].

Desmin connects different cell organelles and multi-protein complexes [20] such as Z-bands [21], costameres [22] and desmosomes [23]. Therefore, desmin filaments have high relevance for the structural integrity of (cardio)myocytes [24]. Mutant desmin disorganizes the complex cytoskeleton network and disrupts sarcomere organization [25].

Desmin consists of three different domains: the N-terminal head, a central rod and a C-terminal tail domain [1]. The central α-helical rod domain consists of two coil subdomains ‘coil-1′ and ‘coil-2′, separated by a non-helical linker L12 [26]. Coil-1 contains a second non-helical linker L1 dividing coil-1 into a smaller 1A and a larger 1B region [26,27]. A coiled-coil dimer is formed by dimerization of the central rod domains [28,29]. These dimers form antiparallel tetramers [27]. The antiparallel tetramers laterally anneal into unit length filaments (ULFs), which are the essential building blocks of desmin filaments [30]. In addition, IFs can fuse end to end, and subunits are intercalary exchanged [31,32]. Pathogenic desmin mutations disturb the complex filament assembly at different stages [33,34], and have a dominant effect on the wild-type form [3]. This explains, at the molecular level, the autosomal dominant inheritance of most *DES* missense mutations.

Currently, it is difficult to predict and classify novel *DES* missense variants in human genetics, since functional data are missing in most cases. Therefore, most of them must be classified as variants of unknown significance (VUSs) according to the guidelines of the American College of Medical Genetics and Genomics (ACMG) [35]. Especially, because only a few pathogenic *DES* mutations—some with a severe clinical phenotype—have been described in the 1A subdomain [3,6,13,36,37,38,39], we addressed the question: which of the known VUSs within the *DES* 1A coil domain affect desmin filament assembly.

To answer this question, we inserted all known VUSs of the 1A subdomain listed in the ClinVar database (https://www.ncbi.nlm.nih.gov/clinvar/, accessed on 21 July 2022) [40] and the Human Gene Mutation Database (https://www.hgmd.cf.ac.uk/, accessed on 21 July 2022) [41] into expression plasmids and performed cell transfection experiments using SW-13 and H9c2 cells, as well as cardiomyocytes derived from human-induced pluripotent stem cells (hiPSCs) in combination with cytochemistry and confocal microscopy. We found that the N-terminal part of the 1A desmin subdomain is a hot spot for variants affecting filament assembly in vitro. Our functional data may contribute to an improved understanding of novel *DES* mutations within this subdomain, which may support variant classification and the genetic counselling of affected cardiomyopathy patients in the future.

## 2. Materials and Methods

### 2.1. Plasmid Generation and Site-Directed Mutagenesis

The cloning of the plasmid pEYFP-N1-DES-WT has previously been described (Appendix A) [34]. All mutations were inserted into this plasmid by site-directed mutagenesis (SDM) using the QuikChange Lightning kit according to the manufacturer’s instruction (Agilent Technologies, Santa Clara, CA, USA). The used primers (synthesized by Microsynth, Balgach, Switzerland) are listed in Appendix B. The *DES*-encoding regions of all plasmids were verified with Sanger sequencing (Macrogen, Amsterdam, The Netherlands) using the CMV_for and EGFP_rev primers (Appendix B). Sequencing data were analyzed using SnapGene software Version 6.1 (GSL Biotech LLC, San Diego, CA, USA). The plasmids were prepared using the Plasmid Miniprep Kit (Thermo Fisher Scientific, Waltham, MA, USA). All plasmids are available from the corresponding author (A.B.).

### 2.2. Cell Culture

SW-13 and H9c2 (ATCC, Manassas, VA, USA) were cultured in Dulbecco’s Modified Eagle Medium (DMEM, Thermo Fisher Scientific) supplemented with 10% fetal calf serum and penicilline/streptomycine at 37 °C and 5% CO_2_. SW-13 cells do not express desmin or any other cytoplasmic IF protein [42], whereas H9c2 cells are cardiac myoblasts with an endogenous desmin expression [43]. These cells were split every three days using trypsin/ethylenediaminetetraacetic acid (Thermo Fisher Scientific). HiPSCs (NP0040-8, UKKi011-A, kindly provided by Dr. Tomo Saric, University of Cologne, Germany) were cultured in Essential E8 Medium (Thermo Fisher Scientific), as previously described [44]. The medium was changed every day. Vitronectin (#A14700, Thermo Fisher Scientific) was used for coating the cell culture dishes. HiPSCs were split using Versene (Thermo Fisher Scientific, 4 min, 37 °C).

### 2.3. Differentiation of Induced Pluripotent Stem Cells into Cardiomyocytes

HiPSCs were differentiated into cardiomyocytes by modulating the Wnt-pathway using CHIR99021 and IWP2, as previously described [45,46], and were used after contraction started at day 10 (Appendix A). The video file was recorded using an Eclipse TE2000-U wide-field microscope (Nikon, Tokyo, Japan).

### 2.4. Cell Transfection

H9c2 and SW-13 cells were split one day before transfection and were cultured in 8-well µSlide chambers (ibidi, Gräfelfing, Germany). Lipofectamin 3000 (Thermo Fisher Scientific) was used according to the manufacturer’s instruction for cell transfection. In total, 250 ng of the plasmid was used per well, and the ratio of Lipofectamin 3000 to plasmid was 3:1.

HiPSC-derived cardiomyocytes were washed with phosphate-buffered saline (PBS) and treated with accutase (Sigma-Aldrich, St. Louis, MO, USA) for 6 min at 37 °C. Afterwards, the cardiomyocytes were resuspended in culture medium and centrifuged at 200x g for 5 min. After resuspension, the cardiomyocytes were cultured in Geltrex-coated µSlide chambers (ibidi). Lipofectamin 3000 (Thermo Fisher Scientific) was used to transfect iPSC-derived cardiomyocytes. The cardiomyocytes were incubated after transfection in µSlide chambers (ibidi) for 24 h.

### 2.5. Cell Fixation and Immunocytochemistry

Twenty-four hours after cell transfection, the cells were gently washed with PBS (Thermo Fisher Scientific). Afterwards, the cells were fixed with 4% HistoFix (Carl Roth, Karlsruhe, Germany) for 15 min at room temperature (RT). Then, the cells were washed twice with PBS and permeabilized using 0.1% Triton X-100 (solved in PBS) for 15 min at RT. After washing with PBS, the SW-13 cells were incubated for 40 min at RT with phalloidin conjugated with Texas-Red (1:400, Thermo Fisher Scientific) to stain the F-actin. iPSC-derived cardiomyocytes were co-stained with primary anti-sarcomeric α-actinin antibodies (1:100, 4 °C, overnight, #A7732, Sigma-Aldrich) in combination with secondary Cy3-conjugated anti-mouse immunoglobuline antibodies (1:100; RT, 1 h, #115-165-068, Jackson ImmunoResearch, Ely, UK). 4′,6-diamidino-2-phenylindole (DAPI, 1 µg/mL) was used for the staining of the nuclei for 5 min at RT. After two final washing steps with PBS, the cells were stored in PBS in the µ-Slide chambers at 4 °C in the dark until microscopy was performed.

### 2.6. Confocal Laser Scanning Microscopy

The TCS SP8 confocal system (Leica Microsystems, Wetzlar, Germany) was used in combination with Application Suite X software (Leica Microsystems) for the confocal fluorescence microscopy of the transfected cells. DAPI was excited at 405 nm and the emission was detected in the range between 410 and 460 nm. Enhanced yellow fluorescent protein (EYFP) was excited at 488 nm and the emission was detected in the range between 493 and 560 nm. Texas Red was excited at 552 nm and the emission was detected in the range between 570 and 781 nm. Cy3 was excited at 552 nm and the emission was detected in the range between 556 and 758 nm. Excitation and emission detection for the three fluorescence channels was sequentially performed. Then, 3D stacks were imaged for hiPSC-derived cardiomyocytes and presented as intensity projections.

### 2.7. Molecular Visualization

The desmin dimer structure was predicted using AlphaFold-Multimer [47] and was visualized using the PyMOL Molecular Graphics System Version 2.5.2 (Schrödinger, New York, NY, USA).

### 2.8. Statistical Analysis

Over one hundred cells were manually analyzed per transfection experiment, and six independent transfection experiments were analyzed per construct and cell type (*n* = 6). A nonparametric Kruskal–Wallis test followed by Dunn’s multiple comparison was performed using GraphPad Prism Version 9.0 (GraphPad Software, San Diego, CA, USA). The presented data are shown as mean ± standard deviation (SD).

## 3. Results

Most variants within the desmin 1A subdomain are currently classified as VUSs. This subdomain is highly conserved in different type-3 intermediate filament proteins, as well as between different species (Figure 1A,B). However, the impact of specific variants is unknown in most cases. To evaluate the impact of 34 different VUSs within the desmin 1A subdomain listed in the ClinVar database (Figure 1C), we inserted them via SDM into expression plasmids and performed cell culture transfection experiments in combination with confocal microscopy. We used SW-13 cells, because this cell line expresses no endogenous desmin or any other cytoplasmic intermediate filament proteins. Additionally, we used the cardiac myoblast cell line H9c2 as well as hiPSC-derived cardiomyocytes for these experiments, because these cells express endogenous desmin. Wild-type desmin forms filamentous structures in all cell types (Figure 1D–F). Representative non-transfected cells are shown in Appendix A.

In contrast to wild-type desmin, the mutants p.E111G, p.L112R, p.L115P, p.N116I, p.N116K, p.R118C and p.R118S, which are localized at the N-terminus of the 1A subdomain, disturb cellular filament formation in vitro and cause a cytoplasmic desmin aggregation in both cell lines, as well as in iPSC-derived cardiomyocytes (Figure 2). The desmin mutants p.E114G and p.D117H form filamentous networks comparable to the wild-type protein (Figure 2C,G).

The desmin mutants p.A120P, p.Y122D, p.Y122C, p.I123N, p.V126L, p.R127G and p.R127P, which are likewise localized in the N-terminal part of the 1A subdomain, cause comparable cytoplasmic desmin aggregates of different sizes (Figure 3 and Figure 4). However, some VUSs localized in the N-terminal part of the highly conserved 1A subdomain (p.D117H, p.N121H, p.I123V, p.E124G/A and p.V126M) form regular intermediate filaments comparable to the wild-type desmin (Figure 3 and Figure 4A). Similarly, no VUSs localized in the C-terminal part of the 1A subdomain (p.L129R, p.Q131K, p.A135V, p.L136V, p.L136H, p.A137D, p.E139Q, p.E139K, p.V140M, p.V140L, p.L143V, p.L143P and p.G145D) interfere with filament formation (Figure 4 and Figure 5). Even drastic amino acid exchanges of a negative to a positive amino acid (p.E139K) do not affect filament formation (Figure 5C). Comparably, the insertion of a proline at position 143 (p.L143P) does not affect desmin filament formation (Figure 5G). The quantification of >100 transfected cells in six repeated independent transfection experiments verified these filament formation defects (Figure 6A and Appendix A). Of note, all missense variants leading to filament assembly defects are localized within the N-terminal part of the 1A desmin subdomain, whereas the C-terminal mutants do not affect filament formation, supporting that this region is a mutation hot spot in the desmin protein (Figure 6B–D).

## 4. Discussion

In this study, we found that several VUSs localized in the N-terminal part of the desmin 1A subdomain affect filament assembly in vitro, leading to abnormal cytoplasmic protein aggregation. Our results indicate that this desmin region is a hot spot for pathogenic mutations, leading to skeletal myopathies or cardiomyopathies.

Different criteria such as, e.g., co-segregation within the family or absence in healthy controls, are frequently used in cardiovascular genetics for pathogenicity classification and risk assessment [35]. However, for most novel rare variants, the pathogenetic impact is unknown and difficult to predict. As a database resource for genetic variants, identified VUSs are collected, e.g., in the ClinVar database (https://www.ncbi.nlm.nih.gov/clinvar/, accessed on 21 July 2022) and/or the Human Gene Mutation Database (https://www.hgmd.cf.ac.uk, accessed on 21 July 2022). According to the ACMG guidelines, functional data are a strong criterion for pathogenicity (PS3), and can support the classification and pathogenetic evidence of specific VUSs [35]. Some pathogenic or likely pathogenic variants in the desmin 1A subdomain have recently been characterized [6,36,38,39,48]. However, variants within the 1A desmin subdomain are less characterized compared to desmin coil-2 or the tail domains [33,49,50]. Therefore, we focused in this study on VUSs distributed over the complete 1A desmin subdomain. We inserted a set of 34 different variants, classified in ClinVar as VUSs, into a desmin-encoding expression plasmid and analyzed their cellular filament assembly via confocal microscopy. These experiments revealed 14 different VUSs (p.E111G, p.L112R, p.L115P, p.N116I, p.N116K, p.R118C, p.R118S, p.A120P, p.Y122D, p.Y122C, p.I123N, p.V126L, p.R127G and p.R127P) and a severe filament assembly defect (Figure 7). These functional data support their classification as ‘likely pathogenic’ variants (ACMG, class 4) rather than VUSs (ACMG, class 3). However, we cannot exlude in our experiment that some of the other VUSs change the nanomolecular properties of desmin filaments, such as those previously described by Kreplak et al., e.g., for desmin-p.Q389P and p.D399Y [51].

Remarkably, there are two positions (p.I123 and p.V126) where only one amino acid exchange causes desmin aggregation (p.I123N and p.V126L) and the other one does not (p.I123V and p.V126M). In the case of p.I123, this might be explained by the exchange of a hydrophobic isoleucine against the polar asparagine residue (p.I123N). In contrast, the exchange against valine at this position introduces a hydrophobic and smaller amino acid (p.I123V). In the case of p.V126L, this might be explained by steric hindrance, since leucine is larger than valine and might interfere with desmin filament assembly. However, according to the ACMG guidelines, a novel missense variant at the same position as a previously described pathogenic mutation is a moderate criterion for pathogenicity (PM5) [35]. On the other hand, the examples of positions p.I123 and p.V126 demonstrate that this criterion should be handled with care in the context of *DES* mutation, because only functional analysis can elucidate the disparities between different missense mutants.

Interestingly, the aggregate causing mutations cluster in the N-terminal part of the 1A desmin subdomain, whereas most of the C-terminal VUSs did not affect the desmin filament formation in the transfected SW-13 or H9c2 cells, or the hiPSC-derived cardiomyocytes. These results indicate that the N-terminal part of the 1A subdomain is a putative hot spot region for pathogenic sequence variants in the desmin protein. In more detail, the amino acids leading to desmin aggregation are oriented to the inner part of the predicted desmin dimer model (Figure 6B,C), whereas mutations affecting amino acids oriented to the outer surface of the dimer, such as p.E114G, p.D117H or p.N121H, have no impact on in vitro filament assembly. It is known that this part of the 1A subdomain of the homologous IF protein vimentin is likewise involved in tetramer formation [28]. However, the detailed molecular structure at the complete IF level is currently unknown. Interestingly, several pathogenic mutations in this part of the 1A subdomain of the homologous protein glial fibrillary acidic protein (GFAP) cause Alexander disease, which belongs to the leukodystrophies (MIM, #203450) [52,53,54,55]. Additionally, mutations in the N-terminal part of the 1A subdomain of lamin A/C (*LMNA*) cause congenital muscular dystrophy or Emery–Dreifuss muscular dystrophy (MIM, #181350) [56,57,58].

## 5. Conclusions

In conclusion, our study revealed a mutational hot spot region in the N-terminal part of the 1A desmin subdomain, where several missense mutations cause severe filament assembly defects. This suggests that novel *DES* missense mutations affecting this part of desmin should be considered as putative disease-causing variants and should be functionally analyzed in the future.

## Figures and Tables

**Figure 1 cells-11-03906-f001:**
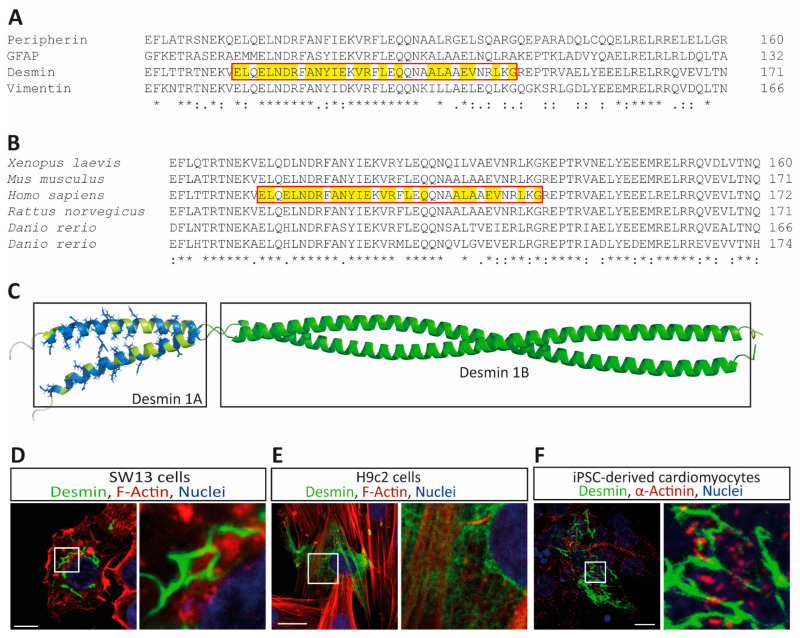
(**A**) Partial sequence alignment of different human type-3 intermediate filament proteins. The highly conserved 1A domain is shown by a red box and the positions of variants with unknown significance are highlighted in yellow. (**B**) Partial sequence alignment of desmin of different species. The highly conserved 1A domain is shown by a red box and the positions of variants with unknown significance are highlighted in yellow. (**C**) Structural overview of the desmin coil 1A and coil 1B subdomains. The positions of variants with unknown significance are highlighted in blue. Representative cell images of desmin wild-type-transfected SW-13 (**D**), H9c2 cells (**E**) and iPSC-derived cardiomyocytes (**F**) are shown. Desmin is shown in green, F-actin or α-actinin in red, and the nuclei are shown in blue. Scale bars represent 10 µm (**D**,**F**) or 20 µm (**E**).

**Figure 2 cells-11-03906-f002:**
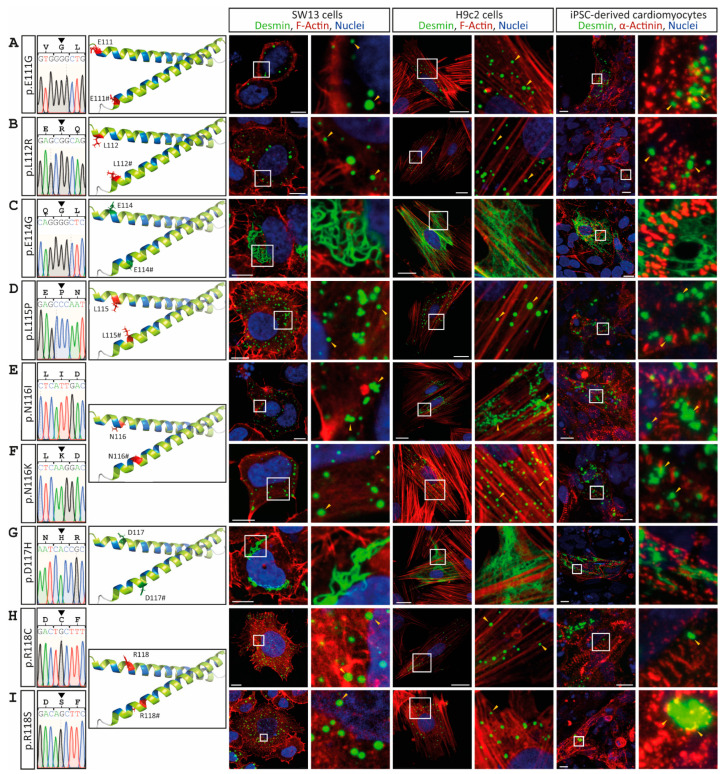
Overview of the *DES* variants (**A**) p.E111G, (**B**) p.L112R, (**C**) p.E114G, (**D**) p.L115P, (**E**) p.N116I, (**F**) p.N116K, (**G**) p.D117H, (**H**) p.R118C and (**I**) p.R118S. Sections of Sanger sequencing electropherograms are shown. The affected amino acids are shown in the molecular overview as sticks in red or green, depending on aggregate or filament formation. Representative cell images are shown. Desmin is shown in green, F-actin or α-actinin in red, and the nuclei in blue. Of note, the mutants p.E111G, p.L112R, p.L115P, p.N116I, p.N116K, p.R118C and p.R118S are forming abnormal cytoplasmic desmin aggregates (yellow arrow heads). Scale bars represent 10 µm (SW-13 cells and iPSC-derived cardiomyocytes) or 20 µm (H9c2 cells).

**Figure 3 cells-11-03906-f003:**
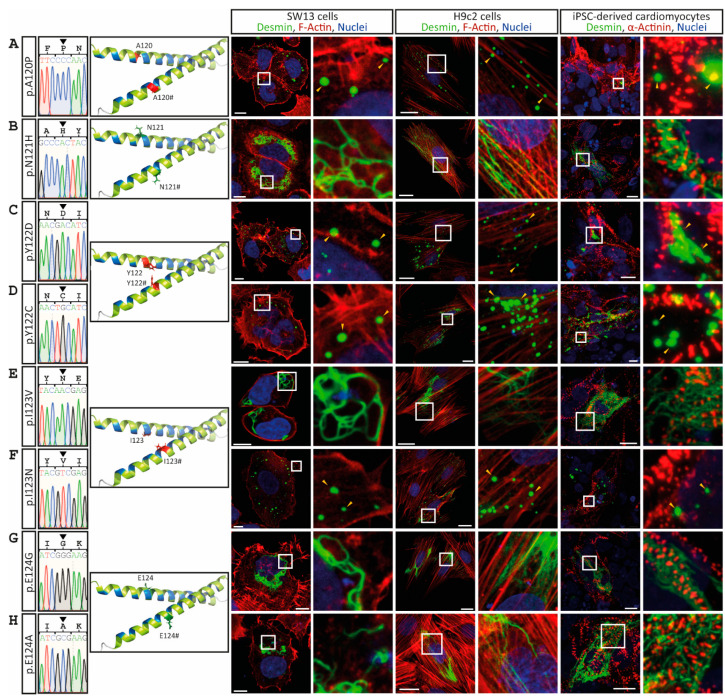
Overview of the *DES* variants (**A**) p.A120P, (**B**) p.N121H, (**C**) p.Y122D, (**D**) p.Y122C, (**E**) p.I123V, (**F**) p.I123N, (**G**) p.E124G and (**H**) p.E124A. Sections of Sanger sequencing electropherograms are shown. The affected amino acids are shown in the molecular overview as sticks in red or green, depending on aggregate or filament formation activity. Representative cell images are shown. Desmin is given in green, F-actin or α-actinin in red, and nuclei in blue. Of note, the mutants p.A120P, p.Y122D, p.Y122C and p.I123N are forming abnormal cytoplasmic desmin aggregates (yellow arrow heads). Scale bars represent 10 µm (SW-13 cells or iPSC-derived cardiomyocytes) or 20 µm (H9c2 cells).

**Figure 4 cells-11-03906-f004:**
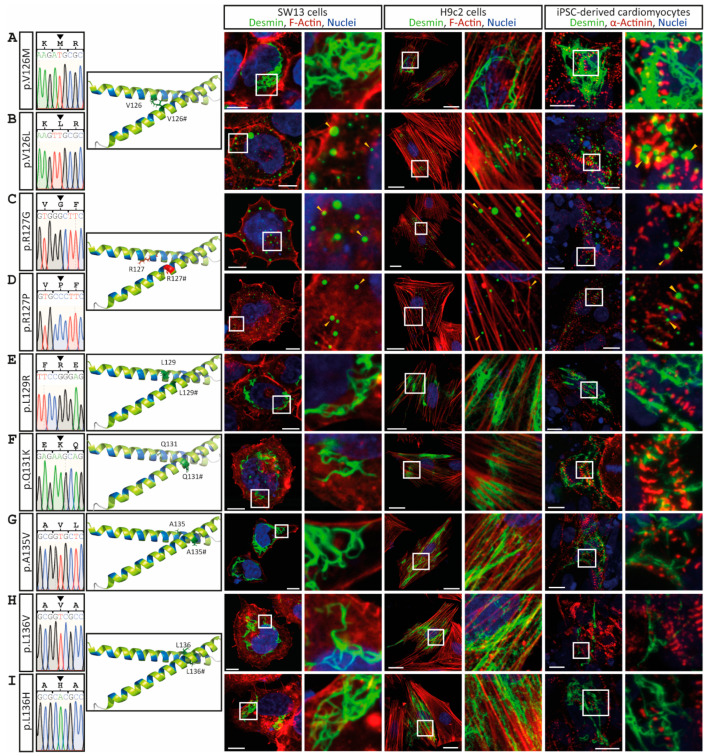
Overview of the *DES* variants (**A**) p.V126M, (**B**) p.V126L, (**C**) p.R127G, (**D**) p.R127P, (**E**) p.L129R, (**F**) p.Q131K, (**G**) p.A135V, (**H**) p.L136V and (**I**) p.L136H. Sections of Sanger sequencing electropherograms are shown. The affected amino acids are shown in the molecular overview as sticks in red or green, depending on aggregate or filament formation. Representative cell images are shown. Desmin is shown in green, F-actin or α-actinin in red, and the nuclei in blue. Of note, the mutants p.V126L, p.R127G and p.R127P form abnormal cytoplasmic desmin aggregates (yellow arrow heads). Scale bars represent 10 µm (SW-13 cells and iPSC-derived cardiomyocytes) or 20 µm (H9c2 cells).

**Figure 5 cells-11-03906-f005:**
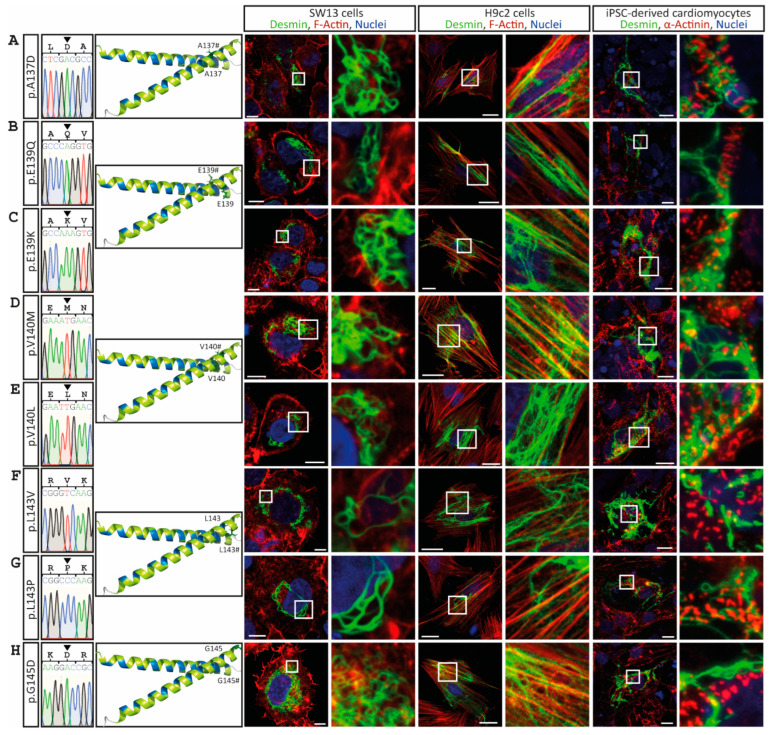
Overview of the *DES* variants (**A**) p.A137D, (**B**) p.E139Q, (**C**) p.E139K, (**D**) p.V140M, (**E**) p.V140L, (**F**) p.L143V, (**G**) p.L143P and (**H**) p.G145D. Sections of Sanger sequencing electropherograms are shown. The affected amino acids are shown in the molecular overview as sticks in red or green, depending on aggregate or filament formation. Representative cell images are shown. Desmin is shown in green, F-actin or α-actinin in red, and nuclei in blue. None of these mutants form abnormal cytoplasmic desmin aggregates. Scale bars represent 10 µm (SW-13 cells and iPSC-derived cardiomyocytes) or 20 µm (H9c2 cells).

**Figure 6 cells-11-03906-f006:**
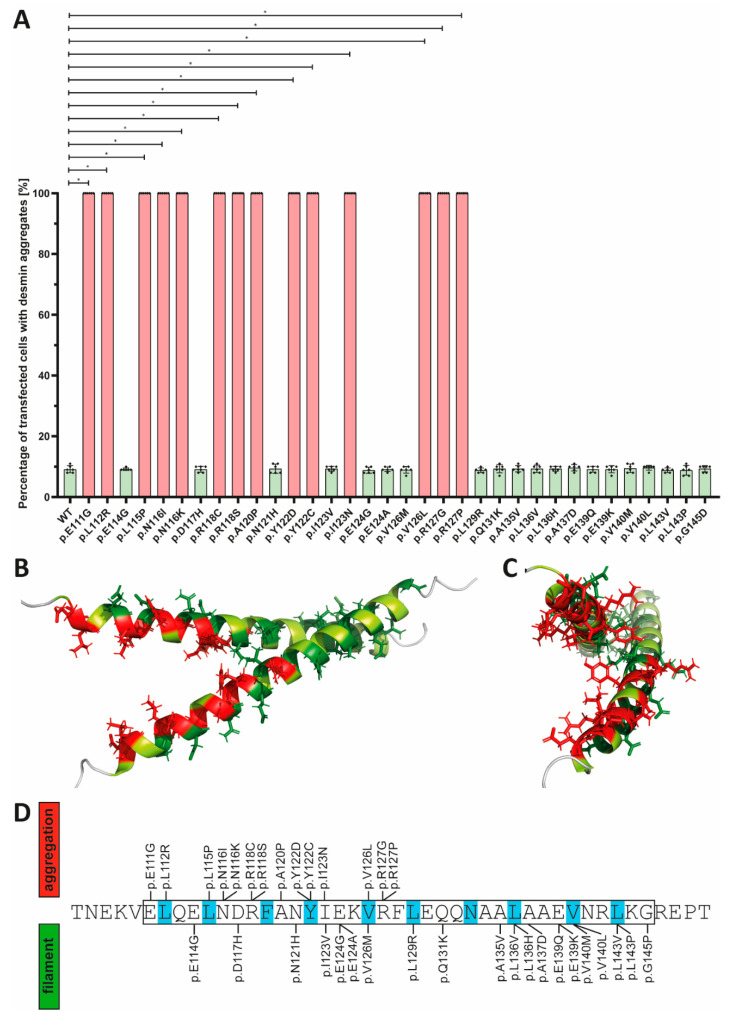
(**A**) Statistical analysis of aggregate formation in transfected SW-13 cells. Kruskal–Wallis test followed by Dunn’s multiple comparison was used for analysis. * *p*-value < 0.05. Error bars represent standard deviation (SD). (**B**,**C**) Structural overview of the desmin 1A subdomain. The amino acid positions where mutant amino acids cause abnormal cytoplasmic desmin aggregation are shown in red, and amino acids where mutants form desmin filaments are shown in green. (**D**) Schematic overview of the localization of the VUSs localized in the desmin 1A domain. The heptad is highlighted in blue.

**Figure 7 cells-11-03906-f007:**
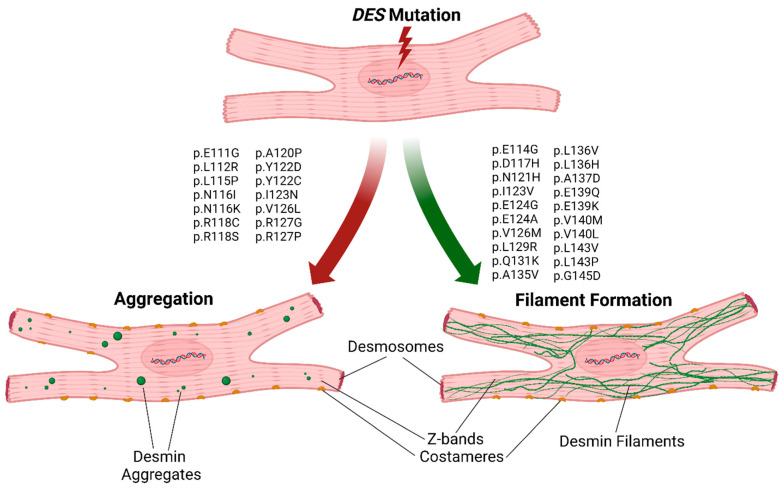
Schematic overview of the aggregate or filament formation of the analyzed desmin mutants. Figure created with BioRender.com.

## Data Availability

The published article and the Appendix A include the data generated or analyzed during this study. The used plasmids are available from the corresponding author (A.B.).

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
