# Peer review of "The N-Terminal Part of the 1A Domain of Desmin Is a Hot Spot Region for Putative Pathogenic DES Mutations Affecting Filament Assembly"

_cells, 2022, doi:10.3390/cells11233906_

Round 1

Reviewer 1 Report (Previous Reviewer 2)

My previous comments have been sufficiently addressed.

There is an error in chapter numbering (2.3 is missing) and the link to CleanVar (line 304) is not correct.

Author Response

We want to thank Reviewer #1 for the constructive peer-review process. We have changed the chapter numbering and also the link to ClinVar in the revised version of the manuscript. 

Reviewer 2 Report (Previous Reviewer 1)

The current manuscript "The N-terminal part of the 1A domain of desmin is a hot spot 2 region for putative pathogenic DES mutations affecting the 3 filament assembly" is clear and presented in a well-structured manner. It was great that the author validated the current findings in human iPSC-derived cardiomyocytes as well. It would be more interesting if the author could find any functional differences in cardiomyocytes transfected with those different DES mutations in their future studies.  However, the current manuscript can be accepted for publication in its present form.

Author Response

We want to thank reviewer #2 for the time and effort to improve our manuscript in a constructive way. Thanks!

This manuscript is a resubmission of an earlier submission. The following is a list of the peer review reports and author responses from that submission.

Round 1

Reviewer 1 Report

Brodehl et al. show that the N-terminal part of the 1A coil domain is a hot spot for pathogenic desmin mutations, which affect the desmin filament assembly. The current study is interesting and relevant to the field of cardiovascular diseases. However, this reviewer has some concerns that need to be addressed before publication.

1. It would be better to add the non-transfected SW13 and H9c2 cell images as a comparison in figure 1D.

2. Abstract (lines 22-23): The consequence of the mutations and desmin aggregations leading to skeletal and/or cardiac myopathies was not proved in the present study. Some in vivo functional study will be required to claim that statement. The author may need to reframe the statements accordingly.

3. Since, desmin filaments have high relevance for the structural integrity of cardiomyocytes (lines 44-45), it would be nice to validate the current findings in the cardiac cells (like human iPSC-derived cardiomyocytes).

4. It would be great if the author could do some functional studies like electrophysiology, calcium handling, and hypertrophy in cardiomyocytes transfected with those different DES mutations.

5. In figure 6A, the sample size for the analysis of desmin aggregate formation is low (n=3). A decent sample size will be required to confirm the current findings.

Reviewer 2 Report

The manuscript "The N-terminal part of the 1A domain of desmin is a hot spot region for putative pathogenic DES mutations affecting the filament assembly" is well written. It addresses a topic of value for the community and the study is well conducted. I only have minor requests for changes before publication.

1) The authors report results for two pars of mutants for the same amino acid (p.I123V & p.I123N and p.V126M & p.V126L) where only one of the mutations of each pair cause aggregation whereas the other behaves like wt. This is not mentioned or discussed in the manuscript. Especially in relation to PM5 of the ACMG guidelines this should be discussed.

2) The text is generally very good, but a few sentences are somewhat strange and should be corrected.

Line 207:  "can be classified in conclusion as likely pathogenic"?

Line 213:  "unknown and hardly to predict". Should probably be "hard"

3) For some reason figure 4 has smaller panels than the others and this should be corrected. It would be nice if 3I and 4A (the p.I126 mutant pair) were in the same figure. One should consider merging figures 2-5 into one as it basically is the same thing

Reviewer 3 Report

The article does not reach enough amonut of data and novelty to be published in a full lenght article